# Nasal Microbiome and Its Interaction with the Host in Childhood Asthma

**DOI:** 10.3390/cells11193155

**Published:** 2022-10-07

**Authors:** Yao Zeng, Jessie Qiaoyi Liang

**Affiliations:** 1Department of Medicine and Therapeutics, Li Ka Shing Institute of Health Sciences, CUHK Shenzhen Research Institute, The Chinese University of Hong Kong, Hong Kong, China; 2Centre for Gut Microbiota Research, Faculty of Medicine, The Chinese University of Hong Kong, Hong Kong, China

**Keywords:** nasal microbiota, dysbiosis, asthma, respiratory infection, host–microbiome interaction, metagenomics, metagenome sequencing, prevention, treatment

## Abstract

Childhood asthma is a major chronic non-communicable disease in infants and children, often triggered by respiratory tract infections. The nasal cavity is a reservoir for a broad variety of commensal microbes and potential pathogens associated with respiratory illnesses including asthma. A healthy nasal microenvironment has protective effects against respiratory tract infections. The first microbial colonisation in the nasal region is initiated immediately after birth. Subsequently, colonisation by nasal microbiota during infancy plays important roles in rapidly establishing immune homeostasis and the development and maturation of the immune system. Dysbiosis of microbiota residing in the mucosal surfaces, such as the nasopharynx and guts, triggers immune modulation, severe infection, and exacerbation events. Nasal microbiome dysbiosis is related to the onset of symptomatic infections. Dynamic interactions between viral infections and the nasal microbiota in early life affect the later development of respiratory infections. In this review, we summarise the existing findings related to nasal microbiota colonisation, dynamic variations, and host–microbiome interactions in childhood health and respiratory illness with a particular examination of asthma. We also discuss our current understanding of biases produced by environmental factors and technical concerns, the importance of standardised research methods, and microbiome modification for the prevention or treatment of childhood asthma. This review lays the groundwork for paying attention to an essential but less emphasized topic and improves the understanding of the overall composition, dynamic changes, and influence of the nasal microbiome associated with childhood asthma.

## 1. Introduction

The human body is an ecosystem consisting of different anatomical niches, such as the respiratory tract, gastrointestinal tract, and skin. Each microbial niche possesses its own physiochemical characteristics and is occupied by a specialised set of microbes [1]. The nasal cavity, nasopharynx, sinuses, and oropharynx shape specific microenvironments in the upper respiratory tract, which is constantly bathed in airflow from the external environment and exposed to atmospheric (physical and chemical) factors, including varying humidity, gases, immunological factors, and organic materials [2]. Nasal mucosa is the first-line defence against airborne pathogens [3]. These and other mucosae in the upper respiratory tract are colonised by specialised resident microbiota, presumably providing resistance to potential pathogens from establishing and disseminating towards the lungs, thereby functioning as gatekeepers to respiratory health [4]. Microbial communities throughout the nasal cavity and the upper respiratory tract typically form a continuum in community structure [3].

Nasal microbiota composition and function influence the susceptibility to respiratory illness, such as asthma [5,6,7], pneumonia [8,9], and chronic obstructive pulmonary disease [10,11]. Respiratory tract infections have also been identified as the most common reason for paediatric sick visits, antibiotic use, and health care expenditures during childhood [12,13,14]. Childhood asthma is the most common serious chronic respiratory disease in infants and children. The global prevalence, morbidity, mortality, and economic burden associated with this disease have sharply increased over the last 40 years [15].

Asthma is characterised by a range of respiratory symptoms including wheezing, coughing, shortness of breath, and chest tightness. The frequency and severity of its symptoms vary widely. Uncontrolled asthma and acute exacerbations may lead to respiratory failure [16]. Causes of asthma appear to be multifactorial but are commonly associated with genetics, living conditions, pathogenic infections of respiratory tract, nutrition, and others [17,18]. However, the mechanisms of asthma exacerbations remain poorly understood. Studies on asthma-related risk factors and pathogenesis would, therefore, aid in the effective prevention, diagnosis, and treatment of this disease.

Nasal microbial communities evolve rapidly during infancy and early childhood. Nasal health-associated commensal organisms (e.g., *Corynebacterium*, *Dolosigranulum*, and *Staphylococcus lugdunensis*) play essential functions in protecting hosts from respiratory tract infections [19,20]. Common nasopharyngeal opportunistic pathogens (e.g., *Haemophilus* (*H.*) *influenzae*, *Streptococcus* (*S.*) *pneumoniae*, and *Staphylococcus* (*S.*) *aureus*) can cause diseases when introduced into a susceptible body site or when hosts are immunologically compromised [21,22]. Nasal microbiota dysbiosis is characterized with altered compositions of bacterial species including increases in the number of pathogens and closely correlates with respiratory disease outcomes and overall health. Nasal microbiota dysbiosis not only significantly contributes to the morbidity of respiratory infections but also increases asthma risks and intensification during childhood [23]. 

The rapidly expanding literature on this topic has focused on the composition and role of the gut microbiome in human health and diseases. However, studies on the nasal microbiome in early life and childhood remain limited, particularly in terms of the effects of nasal microbiota composition and dynamics on respiratory diseases and mechanisms underlying host–microbe interactions. In this article, we review the current literature on early-life nasal microbiota composition with respect to health and childhood asthma. In particular, we provide a summary of the interactions between pathogenic bacteria colonisation and host immune responses and metabolic activities in childhood asthma. 

## 2. Early Colonisation of Nasal Microbiota Predicts the Risk of Subsequent Asthma

### 2.1. Early-Life Nasal Microbiota in Health 

The first two years of life are known as a particular window of health vulnerability. Upon delivery, including vaginal birth and caesarean section, the neonate is exposed for the first time to a wide array of microbes from a variety of sources, including maternal bacteria [24]. Similar to other human body sites, the nasopharynx forms an ecological niche occupied by a variety of microbial species reflective of the delivery mode directly [24,25]. The most drastic changes occur during the first year of life and are probably driven by the maturation of the immune system [24]. Using 16S rDNA (V3–V5) sequencing, Mika et al. [26] identified the five most abundant bacterial families (Moraxellaceae, Streptococcaceae, Corynebacteriaceae, Pasteurellaceae, and Staphylococcaceae) in the nasal swabs of healthy infants in the first year of life, with relative abundances and composition varying with age and seasonal changes. In another study using 16S rDNA (V1–V3) sequencing, the five most highly represented genera, in the baseline nasal filter paper samples of healthy infants less than 6 months of age, were *Corynebacterium*, *Streptococcus*, *Staphylococcus*, *Dolosigranulum*, and *Moraxella* [27]. The formation of nasal microbial communities in early infancy is affected by host and environmental factors (e.g., mode, sex, age, and seasonality), and these factors have been found to correlate with the early onset of respiratory diseases such as acute respiratory infections (e.g., asthma and rhinitis) [28,29,30]. These findings indicate the potential role of early-life nasal microbiota homeostasis in maintaining nasal health.

### 2.2. Early-Life Nasal Microbiota Dysbiosis Associated with Subsequent Asthma

Dysbiosis is defined as an imbalance or disruption of the microbial diversity, and the presence of a “dysbiotic” community in the airways may interact with epithelial and smooth muscle cells and cause asthma [31]. The microbial diversity is affected by various factors, such as drugs, surrounding environmental microorganisms, habitat, nutritional availability, host characteristics (e.g., hygiene, immunity, and genetics), physical factors (e.g., oxygen, pH, and moisture), and other microbial interactions [32]. The dysbiosis may happen in the upper or lower respiratory tract (or both). In the critical first year of life, nasopharyngeal microbiome composition is a determinant for infection spread to lower airways and risk of future asthma development [33]. 

A longitudinal study of Danish neonates (*n* = 321) using a culture-based method identified the pathogenic bacterial species *Moraxella* (*M.*) *catarrhalis*, *S. pneumoniae*, or *H. influenzae* in aspirates from the hypopharyngeal region in 21% (*n* = 66) of asymptomatic neonates at 1 month old [34]. During a 5-year prospective follow-up, the early presence of these pathogenic species increased the risk of wheezing, including a first wheezy episode, persistent wheezing, and hospitalisation for wheezing, as well as for subsequent diagnosis of asthma [34]. According to 16S rDNA (V4) sequencing, another longitudinal study of Australian infants (*n* = 234) demonstrated that the nasopharyngeal niche of most infants was initially colonised with *Staphylococcus* or *Corynebacterium*, which was replaced in a stepwise pattern by *Corynebacterium*, *Alloiococcus*, or *Moraxella* during the critical infancy period (≤12 months) [33]. Infants with acute respiratory infections (ARIs) were less colonised by *Staphylococcus* and *Corynebacterium* and more heavily colonised by *Moraxella*, *Haemophilus*, and *Streptococcus.* On the contrary, healthy infants were more colonised by *Staphylococcus*, *Corynebacterium*, and *Alloiococcus*. Notably, in nasopharyngeal aspirates collected before the first ARIs, a high *Streptococcus* abundance was more frequently detected in infants (≤9 weeks of age) who later displayed wheezing at 5 years of age than those did not. The same trend was also evident for wheezing at 10 years of age, indicating that *Streptococcus* was a potential wheezing-associated factor [33]. Using 16S rDNA (V4) sequencing, Toivonen et al. [35] identified four distinct longitudinal nasal microbiota profiles in the nasal swabs of a birth cohort of 704 children at ages 2, 13, and 24 months. The four microbial profiles were classified as persistent *Moraxella* dominance profile (with high *Dolosigranulum* and low *Streptococcus* and *Staphylococc**us* abundances; 48%), *Streptococcus*-to-*Moraxella* transition profile (13%), persistent *Dolosigranulum* and Corynebacteriaceae dominance profile (24%), and persistent *Moraxella* sparsity profile (with persistently high *Streptococcus* and high *Haemophilu* as well as low *Dolosigranulum* abundances; 14%). The last profile of persistent *Moraxella* sparsity showed a significant association with a higher risk of asthma at age 7. By 16S rDNA (V4) sequencing, Tang et al. [36] showed that a *Staphylococcus*-dominant microbiome in nasopharyngeal mucus samples of infants in the first 6 months of life was associated with an increased risk of recurrent wheezing by age 3 years, persisted asthma throughout childhood, and early onset of allergic sensitization.

A case-control study used 16S rDNA (V3–V6) sequencing to analyse the nasal microbiota colonisation in infants with rhinitis and concomitant wheezing, with rhinitis alone and healthy controls (*n* = 122; ≤18 months) [28]. This study showed that five bacterial families (Corynebacteriaceae, Oxalobacteraceae, Moraxellaceae, Aerococcaceae, and Staphylococcaceae) dominated the nasal microbiota with an average summed abundance of 54.6%. Infants with rhinitis, particularly those who had wheezing concurrently, had nasal microbiome profiles different from those of healthy controls. Higher abundances of Oxalobacteraceae and Aerococcaceae and lower abundances of Corynebacteriaceae and Staphylococcaceae were observed in the rhinitis with wheezing group compared to the healthy control group. The significantly higher abundance of Oxalobacteraceae in patients with wheezing at an early age (up to 9 months) might also have promoted the early colonisation and increased number of *Alloiococcus* species, a common pathogen in otitis media [28]. 

Early nasal colonisation by *Moraxella*, *Haemophilus*, *Streptococcus*, and *Staphylococcus* has been significantly correlated with later chronic wheezing and asthma [33,35,36]. The initial establishment of Oxalobacteraceae might also be related to wheezing disorders and could be a predictive marker of subsequent increases in pathogen such as *Alloiococcus* [28]. Previous research findings imply that specific nasal bacterial exposures early in life can influence the development of asthma.

### 2.3. Early-Life Viral Respiratory Infection on Subsequent Development of Asthma

Respiratory syncytial virus (RSV) and rhinovirus (RV) are common, nearly universal, early-life infections. Infants are more likely to manifest immune responses to RSV and RV due to an immature immune system, predisposing them to possible chronic consequences of these severe infections. Early-life viral respiratory infections have been strongly linked to the development of childhood asthma [37,38]. Although there has been a long-standing debate regarding the causal relationship between infant viral respiratory infections and asthma risk, the type or pattern of nasal bacterial colonisation associated with asthma risk has been identified. Several studies have suggested that bacteria and viruses interact in maintaining health and influencing disease. 

*Lactobacillus* abundance was associated with childhood wheezing illnesses in early life during RSV-ARI. Following RSV-ARI, infants with a high *Lactobacillus* abundance had a significantly reduced risk of wheezing compared to those with a low abundance of *Lactobacillus* at age 2 in a 2-year follow-up [38]. However, this finding may not be generalisable to infants without RSV-ARI as all subjects enrolled in this study were confirmed with RSV-ARI. In another child cohort study, *M. catarrhalis* and *S. pneumoniae* detected by quantitative real-time PCR in weekly nasal samples during RV infection were associated with increased asthma symptoms [39]. In a cohort study of American infants (*n* = 132) using 16S rDNA (V1–V3) sequencing, increased nasal abundances of *Haemophilus*, *Moraxella*, and *Streptococcus* and decreased abundances of *Lactobacillus*, *Staphylococcus*, and *Corynebacterium* were detected in infants with RSV-ARI when compared with healthy infants (average ≤ 6 months) during their first winter viral season [40]. Infants with more frequent RV infections had a lower Shannon diversity index [41], indicating that changes in the nasal microbiota associated with RV infections were characterized by a loss of microbial diversity. Persistent bacterial outgrowth occurred after viral infection, especially for the pathogen *M*. *catarrhalis* [34]. During wheezing illnesses, RV infection and the predominance of *Moraxella* at ages 2 and 3 were indicators of persistent childhood asthma [36]. These studies demonstrate that nasal microbiota colonisation and variations during early-life viral respiratory infection correlate with later respiratory health in childhood.

The distinction between commensal and pathogenic bacteria is often unclear, as some bacteria can be both commensal and opportunistic pathogens [42]. Nevertheless, the above studies have identified that the early presence of specific genera (*Streptococcus*, *Moraxella*, *Haemophilus*, and *Staphylococcus*) in nasal microbiota appears to be commonly associated with the development of respiratory diseases. For example, a high nasopharyngeal abundance of *Streptococcus* in early life is associated with an increased risk of wheezing and asthma in later childhood [33], and the presence of Oxalobacteraceae may serve as a predictor of early-onset wheezing [28]. Viral infection may drive the diversity variations of nasal microbiota and be associated with an increase in pathogens [34,41]. On the other hand, *Lactobacillus* may reduce the risk of subsequent wheezing in infants with a history of RSV-ARI [38]. 

## 3. Impacts of Nasal Microbiota Dysbiosis on the Development and Severity of Childhood Asthma

The alteration in microbiota composition induced by pathological conditions leads to health issues. Previous studies have revealed changes in nasal microbial diversity and density with age and season. The alteration of nasal microbiota with age showed an increase in density and a decrease in diversity within the first year of life in healthy infants [26]. In contrast, another study comparing healthy infants and those with rhinitis showed that nasal microbial diversity increased in the former and decreased in the latter over the first 18 months of life [28]. Contradictory findings among studies may reflect that other contemporaneous life events in early infancy (e.g., changes in feeding patterns, antibiotics use, and childcare attendance) may have synergistic or inhibitory effects on colonisation of particular nasal microbiota.

A higher bacterial richness and specific bacterial profiles with more abundant Gram-negative α-proteobacteria and Gram-positive Bacilli were detected in the nasopharynx of summer-born asymptomatic neonates (1 month old) but not in those born in other seasons [43], indicating that birth season impacts the early colonisation of certain pathogens in the upper airways. Another study reported that peak colonisation occurred in fall/winter for *M. catarrhalis* and in winter/spring for *H. influenzae* via regular cycles of colonisation and clearance in healthy children [44]. The apparent increase in *M. catarrhalis* and *H. influenzae* detection in winter is likely a consequence of increased viral respiratory infections, such as influenza, resulting in increased opportunities for secondary infections by bacterial pathogens. McCauley et al. [45] collected nasal mucus samples from asthmatic children (ages 6 to 17) under steroid treatments during periods of respiratory health or first captured respiratory illness across all seasons. They observed that respiratory illnesses and exacerbations increased from late summer through late fall. In samples with first captured respiratory disease, higher relative abundances of multiple *Moraxella* taxa in spring and several *Staphylococcus* taxa in fall increased the risk of asthma exacerbation in a season-specific manner [45]. Several studies reported that infants with more than two respiratory tract infection episodes per year had an accelerated maturation rate of nasal microbiota [28,38,46], implying that respiratory tract infection may drive deviations from the ”normal” rate of the establishment of nasal microbiota. 

### 3.1. Changes in Nasal Microbiota during Early and Middle Childhoods

The nasopharyngeal microbiome profiles in asthmatic children at 18 months of age were highly different compared to those of adults [47]. The nasopharyngeal microbiome at early childhood was dominated by the same six common genera from 2 months to 5 years, with a noticeable increase in within-sample α-diversity after 2 years of age in both healthy and ARI samples [48]. However, it had not yet reached the level of an adult-like nasopharyngeal microbiome, characterized by a much higher α diversity, lack of *Moraxella* and *Corynebacterium*, and less biomass [48,49]. 

Middle childhood (or the middle and late childhood; ages 6 to 12) is a stage where children move into expanding roles and environments, spending more time in school and other activities away from their family. Shifts in microbiota during this period occur due to changes in lifestyle, development of immunity, growth of bones, and so on [50,51]. The respiratory microbiota of middle-childhood children and adolescents (with ages of 6 to 18) with asthma partly overlapped with those of infants (<2 years) with respiratory infections [33,45,47,48,52]. By culture-based assessments, an earlier study reported that in healthy children (ages 2 to 9 years), the nostrils were enriched in Proteobacteria (*Moraxella*, *Haemophilus*, and *Neisseria*) and Firmicutes (*Streptococcus*, *Dolosigranulum*, *Gemella*, and *Granulicatella*), while in healthy adults, the nostrils were dominated by Actinobacteria (*Corynebacterium*, *Propionibacterium*, and *Turicella*) [53], indicating the striking differences in nasal microbiota composition between children and adults. In contrast, the gut microbiome matures to an adult-like state by 3 years of age [54]. The above findings indicate that the transition of respiratory microbiota communities toward a more adult-like configuration may take place over a longer period of childhood than that of other body compartments such as the gut. This may partly be the reason why many studies investigating the relationships between nasal microbiome and respiratory diseases did not categorize the participants into younger (ages 6 to 12) and older groups (ages 13 to 18) beforehand, even though the subjects recruited may cover a wide age range. However, we should notice that identification of the age of transition to an adult-like nasal microbiota by high-throughput sequencing assessments is needed for future studies, as this is a notable gap in current data.

### 3.2. Impacts of Cross-Sectional Changes in Nasal Microbiota on Asthma

A recent cross-sectional study by Pérez-Losada et al. [52] investigated the association of nasal microbiota with various phenotypes of childhood asthma using 16S (V4) rDNA sequencing. Asthmatic participants (*n* = 168, ages 6 to 18) were classified into three phenotypic clusters according to their clinical characteristics. The most abundant bacterial phylum (Proteobacteria) and pathogenic genus (*Moraxella*) associated with asthma varied significantly across different phenotypic clusters. The lowest abundances of these bacteria were detected in the group with the oldest mean age of asthma onset, mostly consisting of patients at risk of refractory asthma. The highest abundances were detected in the group with the youngest age of asthma onset, involving mainly patients with positive allergic asthma. Intermediate bacterial abundances were detected in the third group, involving the lowest proportion of subjects receiving positive allergy tests and subjects with the best outcomes for post-bronchodilator pulmonary function tests. This group might correspond to the best health outcomes. These findings reveal that the abundances of certain taxa are associated with particular paediatric asthma phenotypes. Asthma phenotypes with high and low abundances of specific bacteria, such as Proteobacteria and *Moraxella*, produced worse health outcomes than intermediate abundances of those bacteria. 

By 16S rDNA (V1–V3) sequencing and metagenome shotgun sequencing, another cross-sectional study compared the composition of airway microbiota among healthy controls (*n* = 31), children with asthma (*n* = 31), and children with asthma in remission (*n* = 30) at around 8 years old [55]. The highest abundance of *Staphylococcus* was observed in the asthma group, and higher abundances of *S**treptococcus* occurred in the two disease groups. These two bacteria might be more broadly negatively associated with lung function and bronchial hyperresponsiveness [55]. Studies have shown that differences in the diversity and abundance of nasal microbiota exist between asthmatic patients and healthy subjects and among different asthmatic phenotypes, indicating that specific microbiota alterations may contribute to asthma development and different clinical outcomes.

### 3.3. Impacts of Longitudinal Alterations of Nasal Microbiota on Asthma 

Longitudinal microbiome studies may help to uncover how shifts in microbiome profiles over time correspond to childhood asthma. Using 16S rDNA (V4) sequencing, a longitudinal study analysed the relationship between nasopharyngeal microbiota and ARIs in 244 children during their first 5 years of life [48]. Six genera (*Moraxella*, *Streptococcus*, *Corynebacterium*, *Alloiococcus*, *Haemophilus*, and *Staphylococcus*) remained dominant in the first five years of life. Although the within-sample α-diversity of the nasopharyngeal microbiota increased after 2 years of age for the remainder of the study period, *Moraxella* or both *Alloiococcus* and *Corynebacterium* appeared to be stable colonisers of the nasopharynx of healthy children. The asymptomatic colonisation of the upper airways by disease-associated taxa (*Streptococcus*, *Haemophilus*, and *Moraxella*) increased the risk of later chronic or transient wheezing (at 5 years old) [48]. This result was consistent with previous research showing that *Streptococcus* colonisation in early life is a risk factor for later childhood wheezing and asthma [33]. 

Changes in nasal microbiota over 5.5 to 6.5 months were studied in 40 asthmatic children (ages 6 to 18 years) using 16S rDNA (V4) sequencing by Pérez-Losada et al. [47]. Five genera (*Moraxella*, *Staphylococcus*, *Streptococcus*, *Haemophilus*, and *Fusobacterium*) dominated the nasopharyngeal core microbiome of these asthmatic children, with *Moraxella* and *Streptococcus* fluctuating more noticeably over time than the remaining genera. 

Another study by McCauley et al. [56] collected weekly nasal secretion samples for 16 weeks from 413 asthmatic children (ages 6 to 17 years). In younger asthmatic children, asthma exacerbation was more strongly associated with *Moraxella* species-dominated microbiota, accompanied by a simultaneous rise in eosinophil activation. The nasal microbiota of children without asthma exacerbations were more likely to be dominated by *Alloiococcus*, *Haemophilus*, *Corynebacterium*, or *Staphylococcus*. Nasal microbiota dominated by *Corynebacterium* and *Staphylococcus*, observed in children who did not show exacerbations, were also associated with a generally decreased risk of respiratory diseases. *Moraxella* and *Staphylococcus* were the most frequently detected genera and exhibited the greatest temporal stability in the upper airways of asthmatic children over a 90-day observation period [56], which contradicted the aforementioned study by Pérez-Losada et al. [47]. Another study also revealed the association between variations in nasal, throat, and stool microbiota and asthma status among asthmatic children (*n* = 56, ages 3 to 17) [57]. *Moraxella*, *Streptococcus*, and *Haemophilus* predominated in all the samples from asthmatic patients under age 11. Within each site, the overall communities could not be distinguished during acute exacerbation and in the recovery phase, while nasal microbiota during these two periods were best associated with most of the clinical features (e.g., IgE level and dust mite allergy) compared with throat and stool microbiota. Nasal microbiota showed a higher diversity during acute exacerbation than the recovery phase, suggesting a more unstable and perturbed state presented during acute exacerbation. During acute exacerbation, the nasal microbiota of allergic children tended to be dominated by *Corynebacterium* and *Dolosigranulum*, which are often considered protective against acute exacerbation and have a positive role when the disease is under control. When moving into recovery, the relative abundance of *Staphylococcus* increased [57]. Hou et al. [58] investigated nasopharyngeal samples longitudinally at six time points (2- to 4-week intervals) from school-age asthmatic (categorizing into asthma exacerbation and stable asthma) and healthy children. They demonstrated that the nasopharyngeal microbiome underwent *Moraxella* expansion during asthma exacerbation and presented a high microbiome resilience (recovery potential) after asthma exacerbation. The relative abundances of *Moraxella* increased while *Corynebacterium*, *Anoxybacillus*, and *Pseudomonas* decreased longitudinally in all asthmatic samples. The longitudinal patterns of nasopharyngeal microbiome diversity and composition did not significantly differ between patients with asthma exacerbation and those with stable asthma. This result suggests that the short-term temporal dynamics of the nasopharyngeal microbiome cannot predict asthma exacerbation. Kloepfer et al. [39] longitudinally investigated nasal microbiota for five consecutive weeks during the peak RV season in autumn in children (ages 4 to 12) with current asthma (*n* = 166) and without asthma (*n* = 142). They found that the presence of RV alongside *S. pneumoniae* increased the frequency of exacerbations of asthma compared to weeks with no RV. 

According to the findings of these longitudinal studies, genera such as *Moraxella* and *Staphylococcus* tended to dominate the nasal microbiota of asthmatic children stably [47,48,56,57]. Although *Moraxella* is a stable coloniser in children at almost all ages, its presence at a young age has been predictive of subsequent asthma exacerbation susceptibility [35,56,58]. In contrast to microbial communities in the throat and gut, nasal microbiota showed more obvious fluctuation during acute exacerbation of asthma [57]. Viral infections combined with bacterial colonisation may additionally increase the severity of respiratory diseases [37,39]. 

Both cross-sectional and longitudinal cohort studies have noted the variations in bacterial phyla and genera distribution in groups of asthmatic patients and healthy subjects or different asthmatic phenotypes. These studies have identified changes in nasal microbiota that may contribute to the development of asthma and the increased severity of asthmatic symptoms. However, studies that observe the variations in nasal microbiota in childhood asthma with larger sample sizes and more extended follow-up periods remain necessary to obtain more generalisable and reliable findings. Moreover, investigations focusing on the relationship between the nasal microbiome and subtypes of asthma could be a stepping stone for more detailed clinical application.

## 4. Host–Microbiome Interactions Associated with Asthma Exacerbation

Host–microbiome interactions are believed to play a substantial role in the development of asthma. The microbiome of the upper airway influences dysbiosis in that of the lower airway, which is the actual site of inflammation in asthma and is difficult to access in population studies. Here, we summarise recent studies on immune response-related gene expression signatures and metabolic pathways associated with asthma activity to better understand the possible roles of nasal microbiome–host interactions in childhood asthma.

### 4.1. Host Immune Responses in Association with Nasal Microbiota Dysbiosis in Asthma

Colonisation of the upper airways by pathogenic bacteria in asymptomatic neonates has immune-stimulating functions [59]. *M. catarrhalis* and *H. influenzae* in the airway mucosal lining fluid induced a mixed T helper cell type 1 (Th1), Th2, and Th17 response with high levels of interleukin-1β (IL-1β), tumour necrosis factor-α (TNF-α), and macrophage inflammatory protein-1β (MIP-1β). *S. aureus* also induced a strong Th17 response with elevated IL-17, indicating that colonisation by pathogenic bacteria can induce species-specific inflammatory immune responses in the airway mucosa and may lead to chronic inflammation and later development of asthma in early childhood. Similarly, another study using whole metagenomic shotgun RNA sequencing reported that *M. catarrhalis* dominated the airway microbiota of asthmatic children [60]. Nasal samples from asthmatic patients (ages 6 to 20) with abundant *M. catarrhalis* demonstrated increased apoptosis signals (TNF and C8orf4) and the expression of inflammation mediators (chemokine ligand 20 (CCL20), IL1A, and interleukin-1 receptor-associated kinase 2 (IRAK2)). In contrast, samples from non-asthmatic individuals did not express this *M. catarrhalis*-related inflammatory response signature. 

IL1A is a pleiotropic cytokine involved in various immune responses, inflammatory processes, and haematopoiesis [61]. The virulence characteristic of microbial adhesion and Proteobacteria abundance were significantly associated with variation in the expression of IL1A in nasal epithelial cells [62]. This effect of increased colonisation by Proteobacteria on IL1A expression demonstrated that nasal microbiome characteristics influence host inflammatory and immune responses during asthma. In the aforementioned study by McCauley et al. [56], *M. catarrhalis* and *S. aureus* were isolated from the nasal secretions of asthmatic children using a culture-based method. The sterile biofilm supernatants of these strains were then used to stimulate airway epithelial cell cultures in vitro. *M*. *catarrhalis* and *S*. *aureus* increased gene expression of the proinflammatory cytokine IL-8, while *M. catarrhalis* also increased expression of the proinflammatory cytokine IL-33 and induced significantly greater epithelial damage by assessing lactate dehydrogenase (LDH) release in sterile biofilm supernatants. These findings indicate that some specific nasal airway microbiota influence mucosal integrity and immunity, and dysbiosis of nasal microbiota in asthma can induce abnormal host immune responses.

In addition, host genetic alterations in the pathogen recognition system of innate immunity may increase the risk of recurrent respiratory infections and asthma [63]. Toll-like receptors (TLRs), such as TLR4, are crucial components of innate immunity by recognizing pathogen-associated molecular patterns. Children with *TLR4* Asp299Gly polymorphism often have inappropriate host immune responses due to impaired TLR4-mediated lipopolysaccharide signalling, which results in decreased lipopolysaccharide-induced IL-12 (p70) and IL-10 responses. *TLR4* Asp299Gly polymorphism subsequently predisposes children to repeated nasopharyngeal colonisation by the pathogens *M. catarrhalis* and *H. influenzae* and an increased risk of asthma [63]. 

Previous research has shown that host immune protection against Pneumococci is normally mediated by IL-17-driven mechanisms in a murine model [64]. After pneumococcal infections, wild-type animals with normal IL-17 expression maintained the normal nasopharyngeal microbiome composition and reduced colonisation by potentially pathogenic Proteobacteria in the nasopharynx. Alternatively, IL-17RA knockout animals had increased Proteobacteria colonisation, decreased *Bacteroidetes*, *Actinobacteria*, and *Acidobacteria* colonisation, as well as reduced overall diversity and evenness of the nasopharyngeal microbiome [64]. Therefore, host immune factors may influence the resistance and resilience to nasal pathogen invasion and further affect the homeostasis of the resident microbiome.

### 4.2. Metabolic Alterations Induced by Nasal Microbiota Dysbiosis in Asthma

Studies have shown that metabolic activities are severely perturbed in asthmatic patients [65,66]. The metabolic capacity of the various member species can be explained or predicted by shifts in microbial community composition. Pérez-Losada et al. [62] showed that in asthmatic and non-asthmatic subjects, metabolism differed mainly due to differences in *Moraxella* gene expression, with *Moraxella* contributing to a great proportion of microbial transcripts. The considerably decreased bacterial functional pathways in subjects with asthma were related to glycerolipid metabolism by metagenomics [67]. Significantly different metabolic pathways were also observed between asthmatic and non-asthmatic adults by analysing the expression of microbiome genes in nasal swabs [68]. Compared to non-asthmatics, asthmatics had the lower relative abundances of lysine degradation, N-glycan biosynthesis, caprolactam degradation, and PPAR signalling pathways and higher relative abundances of pentose phosphate pathway, lipopolysaccharide biosynthesis, flagellar assembly, and bacterial chemotaxis, which were mainly due to the higher gene expression of *Bacteroides caccae*, *Escherichia*
*coli*, *Veillonella parvula*, and *Bifidobacterium longum*. The downregulated and upregulated metabolic pathways in asthmatics can contribute to some key pathogenic features of asthma (e.g., airway inflammation and remodelling), autoimmunity, and allergic inflammation. Abnormal gene expression from different contributing bacteria affects subsequent metabolic activity of the relevant pathways, suggesting that dysbiosis of the nasal microbiome influences the inflammation and severity of asthma.

Microbial configurations in the nasopharynx of asthmatic children reflect not only the cooperative and competitive behaviours among microbial symbionts but also their interactions with host immune responses and metabolism, which could have further ramifications for the later development of respiratory infections. Furthermore, host factors (e.g., genetic alterations and immune factors) impairing the ability to defend against pathogens may influence nasal microbiota colonisation and cause a predisposition for asthma.

## 5. Environmental Factors Causing Biases to Data Distribution

Environmental factors (e.g., birth mode, season of birth, childcare attendance, antibiotic treatment, living area, lifestyle, and parental history) used in the various respiratory microbiota models also play an important role in data distribution. These factors not only exert differential effects on the microorganisms but also on asthma. Some factors can cause obvious biases to data distribution, such as antibiotic use, the presence or absence of a childcare group, and even the presence or absence of a peer group. Two or more antibiotic treatments significantly increased the absolute risk of developing asthma, and the link was mediated, in part, by the longitudinal changes in nasal microbiota [69]. Childcare attendance is correlated with the development of recurrent wheezing and asthma in early childhood [70]. The relative abundances of potentially pathogenic bacteria, including *Haemophilus parainfluenzae*, *Streptococcus sp.*, and *Veillonella sp.*, were lower in skin swabs of children in intervention day-care centres compared with children in standard day-care centres [71], indicating that intervention environments may shape the human commensal microbiota associated with immune regulation. However, another study reported that nasopharyngeal microbiome development was not affected by childcare or antibiotics use in the first year of life [36]. 

Regarding microbiome studies, the presence or absence of a peer group commonly means microbiota are sampled from both a disease group and a control group, or from a single group before and after certain symptoms present themselves (as control and disease groups, respectively). Parallel designs should always include a control group for comparison analysis [72]. Although some microbial communities showed a higher correlation on within-individual analysis than on between-individual analysis [73], they are not static. Microbiota shift or develop naturally over time. For example, variations in microbiota can be mistakenly attributed to the effect of acute infection but not the time-related factor in the absence of a healthy peer group as a control; therefore, bias might be produced. During the identification and recruitment of participants, it is essential to match them on information of medical treatment such as drug intake, antibiotics use, childcare intervention, and the presence or absence of a peer group. The above factors can confound study designs, leading to biases in the microbiota composition and the incidence of spurious associations. 

Moreover, there are probably regional differences in asthma that must be taken into account. Regional differences in the prevalence and severity of asthma symptoms have been reported [74,75]. A gradual transition of human gut microbiota in the gross compositional structure and decrease in diversity may have occurred with the three stages of subsistence—foraging, rural farming, and industrialized urban life [76]. Understanding how geographic factors affect microbial community composition may help to determine a particular microbial community’s geographic exclusivity. The loss of beneficial microbes in present-day urban-industrialized societies, such as *Treponema*, may play an important role in “diseases of civilization” such as asthma, allergies, diabetes, obesity, and inflammatory bowel disease [76]. How geographic variables affect the shifts of specific pathogens of the nasopharynx to increase asthma risk needs to be further explored. Standardised research methodology, full records of investigation for environmental variables and geographic regions (and/or lifestyle), appropriate statistical adjustments in bioinformatics analysis models, and extensive surveys of metadata are all helpful in limiting the effect of confounders and further increase the resolution on members of the microbiota that are truly associated with the disease.

## 6. Technical Concerns for Biases of Research Findings

Although the nasal microbiome in asthma has received rising attention in recent years, studies on early-life asthma face various methodological issues, including choice of sampling approach (e.g., nasal swabs, brushes, lavage, and dry filter papers) and sampling sites (e.g., anterior nares, inferior turbinate, and deep nasopharynx). As a deep nasopharyngeal swab may be uncomfortable and cause coughing or gagging for patients, especially for infants, less invasive nasal swabs or dry filter paper may be more common when sampling younger infants. Some studies have contended that microbial signatures spatially vary (biogeography) across nasal microenvironments [7,77], which has likely contributed to the inconsistency in dominant nasal microbiota compositions among some existing studies. However, studies also have supported the view that relationships between nasopharyngeal microbiota profiles and disease severity are replicated in the anterior nares due to their considerable overlaps in bacterial community composition with the nasopharynx [78,79]. Nasal swab samples are, thus, commonly regarded as the qualified sample type and can be used to detect microbial risk markers for respiratory diseases.

Extraction methodology affects bacterial DNA yield, quality, and composition. Commercial kits can extract DNA of higher purity compared to traditional *phenol*–chloroform DNA extraction, as assessed by A260/A280 and A260/A230 [80]. In kit-based DNA extractions, high-quality microbial DNA can be isolated by all tested commercial kits; however, final DNA yields vary significantly across different DNA extraction kits [80,81]. The percentage of contaminants in each library has a power-law relationship with DNA yield, so a lower yield generally correlates with higher contamination [82]. Moreover, the capacity of different DNA extraction kits to capture various microbial communities varies, thus influencing Shannon α-diversity and Chao1 richness in nasal swab samples [83]. Therefore, DNA extraction methods must be standardised in a given project, and comparison among data from different projects should account for the influence of DNA extraction methods on microbiota profiles.

With respect to 16S rDNA sequencing, the sequencing batch and change of operator do not affect the reproducibility of bacterial diversity profiles, but the choice of 16S rDNA primer pairs is a critical determinant of microbiota profiling [80,84]. Results from hypervariable regions targeting V3–V4 and V4–V5 generally produce more reproducible data than those targeting V1–V3 [80]. Sequencing data based on the V4 primer pair showed a higher α-diversity of the gut microbial community compared to other hypervariable regions of the bacterial 16S rRNA gene such as V1–V2 and V3–V4 [84]. Therefore, the selection of 16S rDNA hypervariable regions for sequencing is crucial in assessing the microbial phylogenetic diversity. Although whole metagenome sequencing is widely applied in gut microbiome studies to produce data of much higher coverage and resolution, it is much less frequently applied in the study of the nasal microbiome, likely due to the limited DNA yield of nasal specimens.

In addition, no all-purpose strategy can guarantee the best result for a given project, though combinations of software, parameters, and databases can be tested for efficacy. Ongoing 16S rDNA sequencing analysis tools include, but are not limited to, Phyloseq, UniFrac, mothur, SPINGO, the Ribosomal Database Project, and QIIME [85,86,87]. The performance of different tools in terms of microbial taxonomy and function often differs. For example, QIIME performs with a high specificity but relatively low sensitivity at determining the genus level [88]. DADA2 offers the best sensitivity at the expense of decreased specificity, and USEARCH-UNOISE3 shows the best balance between resolution and specificity [89].

In summary, the consistency and contradiction between previous studies emphasise the importance of standardised research methods. Analysis biases can be generated from any of the following steps: setting and adjustments of environmental variables for data distribution, sampling, DNA extraction, sequencing, and bioinformatics analysis, which lead to an observed community that is significantly altered from the true underlying microbial composition. The establishment of standardised experimental and analytical protocols for all of these procedures is essential to eliminating analysis biases and producing valid metagenomic data. Whole metagenomics sequencing should be applied for nasal microbiome profiles of higher resolution and to identify pathogenic species or strains that cannot be found by 16S rDNA sequencing.

## 7. Perspectives

### 7.1. Microbiota-Targeted Treatment Strategies for Asthma

Current immunomodulatory treatments for asthma mainly target mechanisms of type 2-related eosinophilic inflammation, such as corticosteroids and antibodies to target specific cytokine mediators (e.g., IL-5). However, these immunomodulatory strategies are not effective for patients without type 2 pathway activation [90]. Although the underlying mechanisms linking bacterial composition and asthma are not fully understood, early identification of children at high risk and modification of airway microbiota for childhood asthma present a new avenue for primary prevention strategies. Probiotics have been used in studies of allergic asthma and rhinitis as an intervention method to modify the gut microbiota [91,92,93]. Probiotics are live microbes that can confer health benefits on the recipient host, including the most commonly used genera *Lactobacillus* and *Bifidobacteria*. Newborns administered with prebiotics and probiotics had a significantly lower incidence of respiratory tract infections than a placebo group [94,95]. In a preclinical investigation, *Lactobacillus rhamnosus* could reduce airway inflammation in a mouse asthma model [96]. However, in a meta-analysis, probiotics containing *Bifidobacterium* and *Lactobacillus* species used in four studies were found to have no positive effects on asthma patients [97]. As long-term follow-up data on already initiated cohorts using probiotics for the primary prevention of allergic disease are limited, whether probiotics are helpful in modifying airway microbiota and reducing the risk of asthma has been inconclusive. The optimal method, time of administration, duration of treatment, and strains of probiotics for use need to be established. 

### 7.2. Gut Microbiota Dysbiosis and Asthma

Although many questions remain, emerging findings have cumulatively emphasised the importance of the nasal microbiome in nasal health and the development of asthma. Moreover, risk factors for asthma are not located solely in the nasal cavity. Children who developed asthma at school age displayed a lower gut microbiome diversity during early infancy compared to non-asthmatic children [98]. The immune system is shaped by the establishment of gut microbiome in early life, which has an important impact on the development of asthma [99]. Despite the unclear link between the gut and airway microbiome, the potential interactions between mucosal tissues of the gut and lung via the gut–lung axis have been confirmed [97]. For example, depletion of the gut microbiota led to more severe bacterial pneumonia, while restoring microbiota in the gut reduced the severity of pneumonia in a mouse model [100]. Remote organs and mucosal and haematological immune systems are all affected by the gut microbiome [101]. The metabolic activity of gut microbiota may affect the risk of asthma. For example, *p*-cresol sulphate, a gut microbial-derived product of L-tyrosine metabolism, can reduce CCL20 produced by airway epithelial cells [102]. 

### 7.3. Nasal Microbiota Dysbiosis and COVID-19

The nasal epithelium is one of the first sites of infection with SARS-CoV-2 [103]. Significant decreases in both the Chao1 index and Shannon index of the nasal/oropharyngeal microbiota have been observed in patients affected by coronavirus disease 2019 (COVID-19) [104]. SARS-CoV-2 depends on the cell surface enzyme angiotensin-converting enzyme 2 (ACE2) for host cell entry. Therefore, downregulating ACE2 expression in the nasal epithelium of the upper airway might reduce transmission and decrease the acquisition of SARS-CoV-2 infection [103,105]. However, whether asthmatic patients or those with other respiratory diseases who have developed nasal microbiome dysbiosis are more susceptible to COVID-19 or have more severe outcomes from COVID-19 requires further study.

Future studies should further focus on the mechanistic roles of the identified nasal taxa in asthma, which will help elucidate microbiome–disease–phenotype associations and improve the prevention and treatment of early-onset asthma. It is also important to investigate the microbiome and other risk factors outside the nasal cavity to fully understand the aetiology of asthma.

## 8. Conclusions

A strong association between nasal microbiota dysbiosis and asthma has emerged. The composition and diversity of microbial communities differ largely among different body sites. Simultaneous investigation of the throat, stool, and nasal microbiota demonstrates the importance and uniqueness of nasal microbiota in association with most clinical features of asthma. Early-life nasal dysbiosis causes susceptibility to asthma and allergic inflammation. Particular pathogens colonising the upper respiratory tract may serve as biomarkers for the subsequent risk of asthma development in childhood.

Based on the above-mentioned findings, we summarise the common microbial communities identified in nasal samples and microbiome dysbiosis associated with asthma (Table 1). Nasal microbiota dynamics in ARIs, especially in asthma, as well as host–microbiome interactions, are illustrated in Figure 1. The growing literature has highlighted the significance of the dysbiotic nasal microbiome to the initiation and development of childhood asthma and subsequent risk of asthma exacerbation.

In order to eliminate the apparent inconsistencies among different studies and provide more precise information on the relationship between microbiome dysbiosis and asthma onset, development, and progression, integrative validation and standardised protocols for microbiome studies are imperative. Moreover, due to the various asthma phenotypes, the importance of individualized treatment of asthma is increasing. Existing animal studies have shown promising results of probiotics in treating asthma, but evidence in humans is still limited. Early identification of children predisposing to asthma and microbiota modification might be a promising strategy to prevent or treat asthma. The microbiome and other risk factors outside the nasal cavity should also be investigated to fully understand asthma aetiology.

## Figures and Tables

**Figure 1 cells-11-03155-f001:**
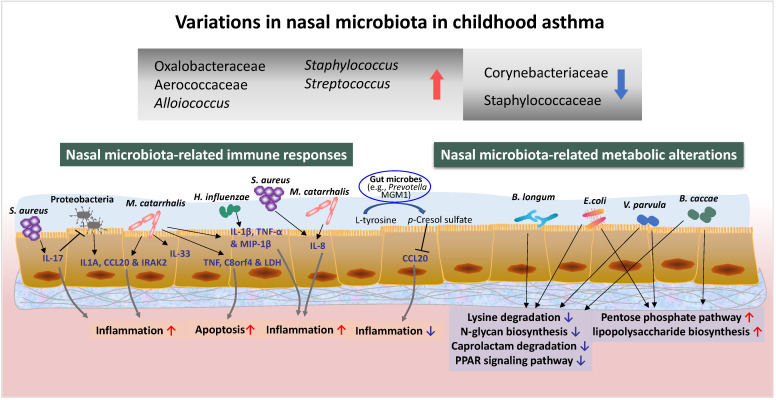
Variations in nasal microbiota and species-specific responses of the airway mucosa and metabolic pathways induced by microbiota colonisation. Changes in representative nasal microbiota, such as higher abundances of Oxalobacteraceae, Aerococcaceae, and Alloiococcus (↑) and lower abundances of Corynebacteriaceae and Staphylococcaceae (↓) [28], are involved in the development of early-onset wheezing in infants. Higher abundances of Staphylococcus and Streptococcus (↑) are more prevalent in asthmatic children than in healthy controls [55]. Opportunistic pathogens that colonise the mucosal layer are associated with host inflammatory immune responses (IL-1β, TNF-α, MIP-1β, IL-17, CCL20, IL1A, IRAK2, IL-8, IL-33), apoptosis signals (TNF and C8orf4), and epithelial damage (LDH) [56,59,60,62]. Host IL-17 signalling can significantly restructure the nasal microbiome and successful resistance to pathogenic Proteobacteria colonisation [64]. *p*-Cresol sulphate, a metabolite of L-tyrosine used by gut microbiota, can protect the host against allergic airway inflammation by reducing CCL20 [102]. The different expression levels of functional genes from specific microbiota (*B. longum*, *E. coli*, *V. parvula*, and *B. caccae*) between asthmatic and non-asthmatic children significantly influence metabolic pathways [68].

**Table 1 cells-11-03155-t001:** Summary of the current findings on the association of nasal microbiota with health and diseases.

Author Year	Population and Initial Status	Sampling	16S rDNA (rRNA Gene) Region	Sequencing Platform	Age Initiation	Design	Key Nasal Microbiota
Mika et al. (2015) [26]	47 healthy infants	Nasal swabs collected biweekly	V3–V5	Roche 454	5 weeks	Longitudinal cohort	Major families in healthy infants: Moraxellaceae, Streptococcaceae, Corynebacteriaceae, Pasteurellaceae, and Staphylococcaceae
Shilts et al. (2016) [27]	33 healthy infants	Nasal filter paper samples	V1–V3	Roche 454	≤6 months	Cross-sectional	Major genera in healthy infants: *Corynebacterium*, *Streptococcus*, *Staphylococcus*, *Dolosigranulum*, and *Moraxella*
Bisgaard et al. (2007) [34]	321 asymptomatic neonates	Hypopharyngeal aspirates	culture-based	N/A	1 month	Longitudinal cohort	Early presence of genera associated with wheezing risk: *M. catarrhalis*, *S. pneumoniae*, and *H. influenzae*
Teo et al. (2015) [33]	234 infants	Nasopharyngeal aspirates at 3 time points (2, 6, and 12 months of age during healthy state) and within 48 h from ARI onset	V4	Illumina MiSeq	2 months	Longitudinal cohort	Nasopharyngeal microbiota was dominated by six common genera: *Moraxella*, *Haemophilus*, *Streptococcus* (more common in ARIs), *Staphylococcus*, *Corynebacterium*, and *Alloiococcus* (more common in healthy samples)
Toivone et al. (2020) [35]	704 children	Nasal and nasopharyngeal swabs collected at 3 time points (2, 13, and 24 months of age during healthy state)	V4	Illumina MiSeq	2 months	Longitudinal cohort	Persistent *Moraxella* sparsity was associated with a significantly higher risk of asthma at age 7
Tang et al. (2020) [36]	285 children	Nasopharyngeal mucus samples at 7 time points (2, 4, 6, 9, 12, 18, and 24 months of age), and during episodes of respiratory illnesses	V4	N/A	2 months	Longitudinal cohort	A *Staphylococcus*-dominant microbiome was associated with increased risk of recurrent wheezing and later asthma development
Ta et al. (2018) [28]	122 infants (60 healthy vs. 62 with respiratory infection)	Nasal swabs at 7 time points (3 weeks and 3, 6, 9, 12, 15, and 18 months of age)	V3–V6	Illumina HiSeq	3 weeks	Longitudinal cohort	Major families in all participants: Corynebacteriaceae, Oxalobacteraceae, Moraxellaceae, Aerococcaceae, and Staphylococcaceae
Rosas-Salazar et al. (2016) [40]	132 infants (33 healthy vs. 99 RSV-infected infants)	Dry filter papers in healthy infants and nasal washes in infants with RSV	V1–V3	Roche 454	average ≤ 6 months	Cross-sectional	Increased *Haemophilus*, *Moraxella*, and *Streptococcus* and decreased *Lactobacillus*, *Staphylococcus*, and *Corynebacterium* in infants with RSV-ARI
Rosas-Salazar et al. (2018) [38]	118 infants with RSV-ARI	Nasal washes	V4	Illumina MiSeq	21.8 weeks	Longitudinal cohort	Decreased genus in infants with recurrent wheezing during RSV-ARI: *Lactobacillus*
Schoos et al. (2020) [43]	328 asymptomatic neonates	Nasopharyngeal swabs	V1–V3	N/A	1 month	Longitudinal cohort	A higher richness and abundance of bacteria (Gram-negative α-proteobacteria and Gram-positive Bacilli) in summer-born asymptomatic neonates
Verhaegh et al. (2011) [44]	1079 healthy children	Nasal swabs at 4 time points (1.5, 6, 14, and 24 months of age)	culture-based	N/A	1.5 months	Longitudinal cohort	Seasonal opportunistic pathogens with peak presence in healthy children: *M. catarrhalis* (autumn/winter) and *H. influenzae* (winter/spring)
McCauley et al. (2022) [45]	208 children with asthma	Nasal mucus samples at two time points during respiratory illness symptoms or asthma exacerbations	V4	Illumina 500	6 years	Longitudinal cohort	Higher relative abundance of *Moraxella* in spring and *Staphylococcus* in fall first captured respiratory illness
Pérez-Losada et al. (2018) [52]	163 children with asthma	Nasal washes	V4	Illumina MiSeq	6–18 years	Cross-sectional	Major pathogenic genera in asthmatic children: *Moraxella*, *Staphylococcus*, *Streptococcus*, and *Haemophilus*
Kim et al. (2018) [55]	92 children (31 healthy vs. 31 with asthma vs. 30 in remission)	Nasopharyngeal swabs	V1–V3/all microbial genomes	Roche 454/Illumina HiSeq 2500	8 years	Cross-sectional	Most abundant genus in asthma group: *Staphylococcus*
Teo et al. (2018) [48]	244 children at high risk of allergic sensitisation	Nasopharyngeal aspirates and blood samples	V4	Illumina MiSeq	2 months	Longitudinal cohort	Stable genera: *Moraxella* or both *Alloiococcus* and *Corynebacterium*
Perez-Losada et al. (2017) [47]	40 children with asthma	Nasopharyngeal washes collected 5.5 to 6.5 months apart	V4	Illumina MiSeq	6 years	Longitudinal cohort	Nasopharyngeal core microbiome of asthmatic children: *Moraxella*, *Staphylococcus*, *Streptococcus*, *Haemophilus*, and *Fusobacterium*
McCauley et al. (2019) [56]	413 children with asthma	Nasal secretion samples collected every 2 weeks throughout the 90-day fall outcome periods	V4	Illumina NextSeq 500	6 years	Longitudinal cohort	Stable species-dominated microbiota in the upper airways of asthmatic children: *Moraxella* and *Staphylococcus*
Liu et al. (2021) [57]	56 children with asthma	Nasal, throat, and gut samples collected. Nasal swabs collected during acute exacerbation and in the recovery phase	V3-V4	Illumina MiSeq	3 years	Longitudinal cohort	Major genera in asthmatic children: *Moraxella*, *Streptococcus*, and *Haemophilus*
Hou et al. (2022) [58]	53 children (20 healthy vs. 33 asthmatic)	Nasopharyngeal swabs of asthmatics at six time points (2- to 4-week intervals) and healthy controls at recruitment	V4	Illumina HiSeq 2500	6 years	Longitudinal cohort	In all asthmatic samples: *Moraxella* increased and *Corynebacterium*, *Anoxybacillus*, and *Pseudomonas* decreased
Kloepfer et al. (2014) [39]	308 children (142 healthy vs. 166 asthmatic)	Nasal mucus samples at five consecutive weeks during a peak RV season	N/A (*Spn9802*, *P6*, and *copB*)	N/A	4 years	Longitudinal cohort	Species inducing asthma exacerbations: *S. pneumoniae* or *M. catarrhalis* accompanying RV infection
Folsgaard et al. (2013) [59]	662 healthy infants	Hypopharyngeal aspirate and airway mucosal lining fluid by filter papers	culture-based	N/A	≤1 month	Cross-sectional	Major species associated with an inflammatory immune response of the airway mucosa: *M. catarrhalis* and *H. influenza*
Castro-Nallar et al. (2015) [60]	14 participants (8 asthmatic vs. 6 healthy)	Nasal epithelial cells	all microbial genomes	Illumina HiSeq 2500	6–20 years	Cross-sectional	Major species in asthma: *M. catarrhalis* (associated with a specific host gene expression signature)
Fazlollahi et al. (2018) [67]	72 adults (21 healthy vs. 51 asthmatic)	Nasal swabs	V3–V4	Illumina MiSeq	10–73 years	Cross-sectional	Major genera in asthmatic adults: *Prevotella*, *Alkanindiges*, *Gardnerella*, and *Dialister*
Lee et al. (2019) [68]	80 adults (60 asthmatic vs. 20 non-asthmatic)	Nasopharyngeal swabs	V1–V3/all microbial genomes	Roche 454/Illumina Hiseq 2500	18–45 years; ≥65 years	Cross-sectional	Major species associated with dysregulated metabolic pathways in asthmatic patients: *Bacteroides caccae*, *Escherichia coli*, *Veillonella parvula*, and *Bifidobacterium longum*

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
