# Peer review of "Nasal Microbiome and Its Interaction with the Host in Childhood Asthma"

_cells, 2022, doi:10.3390/cells11193155_

Round 1

Reviewer 1 Report

Major comments:

The abstract is not informative; it must be improved

Line: 19-21 the manuscript failed to convey the statement

The subsection on microbial changes during mild childhood (ages 3 to 11) is needed.

The paper could be effective by adding more vital information regarding children's nasal microbiota, dysbiosis and asthma. The author may include recent studies like https://pubmed.ncbi.nlm.nih.gov/35889124/ in the manuscript.

Add the information on treatment strategies as a separate sub-heading.

What are all the causes of Early-life nasal microbiota dysbiosis?

Does the gut microbiome have a role in asthma? Explain and connect with the nasal microbiome.

Table 1 has data from all age groups, and they are healthy. Explain?

The conclusion can be improved or rewritten.

References are not in the journal format.

Other comments

Line 95: Change “to” as “in” maintaining nasal health.

Line 111- 112: Rewrite the sentence “Notably, high Streptococcus abundance in healthy 111 nasopharyngeal samples preceding the first ARIs was more frequent in infants (≤ 9 weeks 112 of age)”.

Line 149-153: Merge or rewrite the sentence “infants with significantly lower Lactobacillus abundance following RSV-ARI had a higher risk of wheezing, whereas those with increased abundance of Lactobacillus following RSV-ARI showed a reduced risk of later wheezing illnesses. Lower Lactobacillus abundance was also associated with childhood wheezing illnesses at age 2 years during RSV-ARI [34].”

Line 167-168: check the meaning of lines “Alternatively, Lactobacillus may reduce the risk of subsequent wheezing in infants with a history of RSV-ARI [34]”. Which seems to be controversial with lines 149-153.

Line 241- 245: Rewrite the lines “Those without asthma exacerbations had microbiotas that were more likely to be dominated by Alloiococcus, Haemophilus, Corynebacterium, or Staphylococcus. Corynebacterium and Staphylococcus dominated microbiotas in children who did not exacerbations were also associated with a generally decreased risk of respiratory diseases.”

Add a reference to the lines “Although Moraxella is a stable colonizer in children at almost all ages, its presence at a young age has been predictive of subsequent asthma exacerbation susceptibility. Viral infections combined with bacterial colonization may additionally increase the severity of respiratory diseases.

Abbreviate IL-1β, TNF-α, CCL20, IRAK and PCS while first using the words in the manuscript.

Line 284: Please mention the age of asthmatic patients.

The perspective part might be before the conclusion of the manuscript.

Reviewer 2 Report

The review by Zeng and Liang on the interaction of the nasal microbiome and childhood asthma is a comprehensive, well written manuscript that addresses an important and actively researched interaction between microorganisms in the respiratory tract and development of asthma in children. Although I am not an expert in the field of nasal microbiome or in the field asthma, the manuscript gives a clear insight in the status of the interaction of these fields.

A few remarks:

-          Line 22: the authors claim that they will provide “new insights into prevention and therapeutic strategies”. I haven’t observed these new insights in the manuscript.

-          Line 55: this paragraph is unclear, please rephrase using shorter sentences.

-          Lines 95-96: this sentence is too conclusive as it suggests causality whereas only associations have been described in the paragraph above.

-          Lines 119-120: “explained variance” is in the microbiome field mostly used as a result of a PERMANOVA. Here, I guess, the authors refer to “summed abundances”. Please rephrase.

-          Line 359, Table 1: it would be convenient to have the first author’s name and the year of publication in the table.

-          Line 337 onwards, Perspectives: I like these technical paragraphs, especially the first one on the swabs and the different nasal niches. But these are not perspectives and I would like to see this technical part as the first chapter after the introduction. Then this Perspectives chapter can be used to supply the “new insights into prevention and therapeutic strategies” as promised in the abstract.

Reviewer 3 Report

This review was written after researching many papers and is well written. However, the content is lacking in novelty. I think the problem is that there is too much content and not enough distinction between what is important and what is not important.

Since this is a review paper, it is difficult to comment on individual

data. This is because the data are taken from other papers.

I think the authors' coverage of the literature is fine.

The strength of this paper is that it summarizes and reviews a great

deal of data on the important topic of the relationship between

childhood asthma and infection.

But I must also point out the weak point.

The microbial situation varies greatly depending on the infection at

any given time. Therefore, it is difficult to discuss when conditions

are not consistent when comparing many papers.

The data must be thoroughly examined for each paper to determine under

what circumstances and with what specimens collected.

For example, it must be made clear whether the patient was tested in

the presence of symptoms of acute infection or in the absence of

symptoms.

Another problem is that many articles are reviewed only in relation to

the relationship between asthma and microorganisms, but more attention

should be paid to the many biases.

For example, both asthma and the bacterial infection situation could

be the result of other reasons, and the infection may not have

affected the asthma.

There are many factors that affect asthma, and those factors can also

affect microorganisms. I believe that these other factors act as

obvious biases. For example, the presence or absence of group child

care, antibiotic use, and even the presence or absence of a peer group

may influence this. In addition, there are probably many regional

differences in asthma that must be taken into account.

The authors should note how these issues were addressed in the papers

used for the review and should be discussed in this review.

Reviewer 4 Report

Zeng and Liang analyzed the nasal microbiome and its interaction with the host in children's asthma from various angles regarding early colonization, impacts of nasal microbiota dysbiosis, and host-microbiome interactions. This study laid the groundwork for paying attention to an essential topic while being less emphasized. In particular, it would be a timely study to summarize the studies conducted so far along with the development of research technologies and to lead to future research prospects and clinical applications.

Asthma has a variety of phenotypes, and the importance of individualized treatment is increasing. Studies on the relationship between the nasal microbiome and subtypes of asthma will also be more exciting reviews and could be a stepping stone for more detailed clinical application.

Round 2

Reviewer 1 Report

kindly check for the references section, additional numbering is there.